# Molecular Characteristics, Sources, and Formation Pathways of Organosulfur Compounds in Ambient Aerosol in Guangzhou, South China

Hongxing Jiang[1,2,3,6], Jun Li[1,3], Jiao Tang[1,3], Min Cui[4], Shizhen Zhao[1,3], Yangzhi, Mo[1,3], Chongguo Tian[5], Xiangyun Zhang[1,3], Bin Jiang[1,3], Yuhong Liao[1,3], Yinjun Chen[2], Gan Zhang[1,3]

[1]State Key Laboratory of Organic Geochemistry, Guangdong province Key Laboratory of Environmental Protection and Resources Utilization, and Guangdong-Hong Kong-Macao Joint Laboratory for Environmental Pollution and Control, Guangzhou Institute of Geochemistry, Chinese Academy of Sciences, Guangzhou, 510640, China
[2]Shanghai Key Laboratory of Atmospheric Particle Pollution and Prevention (LAP3), Department of Environmental Science and Engineering, Fudan University, Shanghai 200433, China
[3]CAS Center for Excellence in Deep Earth Science, Guangzhou, 510640, China
[4]College of Environmental Science and Engineering, Yangzhou university, 225009, Yangzhou, China
[5]Key Laboratory of Coastal Environmental Processes and Ecological Remediation, Yantai Institute of Coastal Zone Research, Chinese Academy of Sciences, Yantai, 264003, China
[6]University of Chinese Academy of Sciences, Beijing, 100049, China

*Correspondence to*: Jun Li (junli@gig.ac.cn)

**Abstract.** Organosulfur compounds (OrgSs), especially organosulfates, have been widely reported to be present at large quantities in particulate organic matter found in various atmospheric environments. Despite hundreds of organosulfates and their formation mechanisms being previously identified, a large fraction of OrgSs remain unexplained at the molecular level, and a better understanding of their formation pathways and critical environmental parameters is required to explain the variations in their concentrations. In this study, the abundance and molecular composition of OrgSs in fine particulate samples collected in Guangzhou was reported. The results revealed that the ratio of the annual average mass of organic sulfur to total particulate sulfur was $33\pm12\%$, and organic sulfur had positive correlations with $SO_2$ ($r=0.37$, $p<0.05$) and oxidant ($NO_x+O_3$, $r=0.40$, $p<0.01$). A Fourier transform ion cyclotron resonance mass spectrometry (FT-ICR MS) analysis revealed that more than 80% formular number of the OrgSs detected in the samples had the elemental composition of $O/(4S+3N) \geq 1$, indicating that they were largely in the form of oxidized organosulfates or nitrooxy organosulfates. Many OrgSs that were previously tentatively identified as having biogenic or anthropogenic origins, were also present in freshly emitted aerosols derived from combustion sources. The results indicated that the formation of OrgSs through an epoxide intermediate pathway could account for up to 46% number of OrgSs from an upper bound estimation, and the oxidant levels could explain 20% of the variation in the mass of organic sulfur. The analysis of our large dataset of FT-ICR MS results suggested that relative humidity, oxidation of biogenic volatile organic compounds via ozonolysis, and $NO_x$-related nitrooxy organosulfate formation were the major reasons for the molecular variation of OrgSs, possibly highlighting the importance of the acid-catalyzed ring-opening of epoxides, oxidation processes, and heterogeneous reactions involving either the uptake of $SO_2$ or the heterogeneous oxidation of particulate organosulfates into additional unrecognized OrgSs.

# 1 Introduction

Organosulfur compounds (OrgSs) have been widely identified in atmospheric media including fog, rainwater, and ambient aerosols, and account for a substantial fraction of ambient organic matter mass, with percentage as large as 50% (Surratt et al., 2007; Altieri et al., 2009; Mazzoleni et al., 2010; Luk´Acs et al., 2009; Tolocka and Turpin, 2012; Surratt et al., 2008), which potentially have adverse effects on the global climate system and toxicity to human health (Jimenez et al., 2009; Noziere et al., 2015; Nozière et al., 2010; Nguyen et al., 2012; Bates et al., 2019; Daellenbach et al., 2020). OrgSs is a class of relatively stable and long-lived organic compounds (Olson et al., 2011; Bruggemann et al., 2020), including not only organosulfates (OSs), but also sulfoxides, sulfonates, and sulfones, with OSs identified as the most abundant class (Olson et al., 2011; Chen et al., 2020; Tolocka and Turpin, 2012). A series of studies have reported the hygroscopicity (Peng et al., 2021), light absorption properties (Nguyen et al., 2012; Fleming et al., 2019), and possibly potential toxicity (Lin et al., 2016) of OSs, further highlighting the importance of studying the sources and formation mechanisms of OrgSs.

Various mechanistic studies have revealed the possible reaction pathways by which OSs form. The acid-catalyzed ring opening of epoxides in the presence of sulfuric acid seeds has been widely adopted to explain the formation of OSs from isoprene and other volatile organic compounds (VOCs) (Eddingsaas et al., 2010; Iinuma et al., 2007a; Lin et al., 2013; Bruggemann et al., 2020; Surratt et al., 2010; Lin et al., 2012). Furthermore, heterogeneous reactions between $SO_2$ and unsaturated compounds or aerosol-phase organic peroxides were also identified to generate OSs both by simulation experiments and field observations (Shang et al., 2016; Passananti et al., 2016; Ye et al., 2018; Zhu et al., 2019). Other mechanisms such as nucleophilic substitution of organic nitrates by sulfate (Surratt et al., 2007; Iinuma et al., 2007b; Surratt et al., 2008), sulfate esterification of alcohols/epoxides (He et al., 2014), and sulfoxy radical-initiated oxidation of unsaturated compounds (Nozière et al., 2010; Huang et al., 2019; Wach et al., 2019; Huang et al., 2020) have also been proposed in many studies. Night-time $NO_3$-initiated oxidation of VOCs is considered as an important formation mechanism of nitrooxy-organosulfates (NOSs) (Iinuma et al., 2007b; Bruggemann et al., 2020). The presently proposed formation pathways presumably explain the large variety and ubiquity of OSs; and the above mechanisms suggest that OSs distributions can depend on both precursors of VOCs and inorganic gas (e.g., $SO_2$, $NO_x$, $NH_3$) concentrations, as well as environmental conditions, such as relative humidity (RH), aerosol acidity and oxidant concentrations. However, the OrgSs composition in the actual atmosphere is complex, and many present studies focused on the existing known OSs because they were abundant in particles(Ye et al., 2020; Hettiyadura et al., 2019; Hettiyadura et al., 2017; Wang et al., 2018). A recent study showed that there is a large fraction of OrgSs (67–79%) remaining unexplained at molecular level other than the OSs with known precursors (Chen et al., 2021). Additionally, recent analysis of high-resolution mass spectrometry data showed that OrgSs detected in fresh-emitted sources samples, particularly coal combustion aerosols (Song et al., 2018; Cui et al., 2019; Tang et al., 2020), have a similar molecular composition to classical OSs, complicating the source apportionment and discrimination of reaction mechanisms of OrgSs in the real atmosphere. The above works suggest that there is insufficient

understanding of the comprehensive sources, formation mechanisms and influencing factors of OrgSs for ambient samples (Bruggemann et al., 2020), which makes it an urgent need to fully understand their molecular composition.

Guangzhou is a megacity in South China where featured high temperature, RH and oxidation levels throughout the year, and it is heavily influenced by biogenic−anthropogenic interactions. Studies have shown that Guangzhou often suffers haze events influenced by biomass burning and fossil fuel combustion (mainly vehicle emissions), and organic aerosols can account for large fractions of the total $PM_{2.5}$ in haze (Jiang et al., 2021b; Dai et al., 2015; Liu et al., 2014). Additionally, the high emissions of anthropogenic pollutants (e.g., $NO_x$ and $SO_2$) and high concentrations of particle-phase nitrates and sulfates make the particles very acidic (He et al., 2014). Although several studies have reported the concentrations and possible formation mechanisms of biogenic VOCs (BVOCs) derived OSs in the Pearl River Delta region (PRD) (Bryant et al., 2021; He et al., 2014), these OSs only represented a small fraction of organic aerosol mass. Therefore, a better understanding of the chemical composition, source and influencing factors of OrgSs in Guangzhou will be important to know the particulate pollution and decrease the concentration of secondary organic aerosol (SOA). It will also have important significance for areas where show high temperature, humidity and oxidation levels, and frequent occurrence of secondary processes.

In this study, the molecular composition of atmospheric OrgSs over an urban site in Guangzhou, was characterized by negative electrospray ionization Fourier transform ion cyclotron resonance mass spectrometry (ESI-FT-ICR MS) analysis through accurate mass measurements. The applications of high-resolution FT-ICR MS or Orbitrap mass spectrometry coupled with ESI in studying atmospheric OrgSs have qualitatively provided more new molecular information on OrgSs composition (Ye et al., 2020; Kuang et al., 2016; Lin et al., 2012; Gao and Zhu, 2021). Moreover, FT-ICR MS results combined with chemical tracers and meteorological data were used to evaluate the possible formation pathways and driving factors of OrgSs. We showed that acid-catalysed ring-open of epoxides, heterogeneous reactions of the $SO_2$ uptake pathway and different oxidation processes, were potentially important formation pathways of OrgSs in Guangzhou where usually has high RH, oxidation levels and acidity. This is consistent with a recent field observation that gas-phase oxidation and heterogeneous/multiphase reactions play important roles in SOA formation in Guangzhou (Guo et al., 2020).

## 2 Experimental methods

### 2.1 Collection of $PM_{2.5}$ samples and sulfur-containing species analysis.

A total of 55 atmospheric $PM_{2.5}$ samples (24h) which were collected at an urban site in Guangzhou from July, 2017 to June, 2018, were used for organosulfur analysis. Detailed information about the samples and the measurement of organic tracers, water-soluble inorganic ions and meteorological parameters (including trace gases, temperature, and RH), were described in our recent studies (Jiang et al., 2021b; Jiang et al., 2021a) and in the Supplementary text. Our previous source apportionment using the [14]C-based positive matrix factorization analysis have shown that the primary sources of fossil-fuel combustion and biomass burning averagely contributed half of organic matters at Guangzhou in total, and the rest of organic matters were

associated with secondary processes. It should be noted that the mixed secondary factor of isoprene-derived SOA and organic sulfates formations accounted for 44% of the secondary sources, and showed lower concentrations in winter than in summer (Supplementary text) (Jiang et al., 2021b).

Here, the total fine particulate sulfur (TS) was measured by elemental analyser (Elemental, Germany) and directly compared to inorganic sulfate measured by ion chromatography (IC), and the TS to sulfate-sulfur ratios were calculated (Chen et al., 2021; Shakya and Peltier, 2013; Tolocka and Turpin, 2012). Detailed descriptions of the analysis procedures are presented in the Supplementary text. As assumed, if particulate sulfur was present only as $SO_4^{2-}$, the calculated ratio often shifts from 1 to the small range of 0.9–1.1 using an error propagation method (Shakya and Peltier, 2015, 2013). And the TS to sulfate-sulfur ratios of samples greater than 2 or less than 0.5 were considered as a measure of gross measurement error (Shakya and Peltier, 2015). In this study, the samples' data met to this criterion were excluded from further discussion. Moreover, according to Chen et al. (2021), a calculated ratio of organic sulfur to TS (Org-S/TS) greater than their uncertainty ($\delta_{OrgS/TS}$) is considered significant (detailed calculations can be found in the Supplementary text). The content of organic sulfur (Org-S) was estimated as the amount of sulfate-sulfur subtracted from TS (two negative Org-S values were set as zero). By using this criterion, we exclude the unreasonable data caused by analytical uncertainties associated with measurements. Finally, the concentration data of sulfur-containing species of 40 samples were reserved and used for further discussion.

**2.2 FT-ICR MS analysis on organosulfur compounds**

The feasibility of the method is based on its high mass resolution in identifying mass peaks in conjunction with the assignment of formulas using narrow mass tolerance (< 1ppm absolute mass error for FT-ICR MS results). Previous studies have indicated that the OSs are readily ionized in negative ESI mode, and most of them were observed only in negative mode (Lin et al., 2012; Kuang et al., 2016). All the total 55 $PM_{2.5}$ samples were used for negative ESI-FT-ICR MS analysis and each sample was ultrasonic extracted with methanol in a cold-water bath (Jiang et al., 2021a), because previous studies have suggested that methanol could extracted more than 90% of organic matter both for filed samples and fresh biomass burning samples (Chen and Bond, 2010; Cheng et al., 2017; Huang et al., 2018b). The methanol extracts were filtered with PTFE membranes, concentrated, and directly injected into a 9.4T solariX XR FT-ICR mass spectrometer (Bruker Daltonik GmbH, Bremen, Germany) in negative ESI modes at a flow rate of 180 μL h$^{-1}$ (Jiang et al., 2021a; Jiang et al., 2020). Detailed operating conditions are presented in the Supplementary text. The mass range was set as150–800 Da, and a total of 128 continuous 4M data FT-ICR transients were co-added to enhance the signal-to-noise ratio and dynamic range. Field blank filters were processed and analysed following the same procedures to detect possible contaminations, and all the contaminations in field blanks were subtracted from samples. It should be noted that the general molecular characteristics of samples and their molecular linkages to light absorption properties were reported in our previous study (Jiang et al., 2021a). Here, we focused on the detailed composition of OrgSs and their influencing factors and potential formation mechanisms.

## 2.3 Data processing and statistical analysis

A custom software was used to calculate all mathematically possible formulas for all ions with a signal-to-noise ratio above 4 using a mass tolerance of ±1 ppm. The compounds assigned as $C_cH_hO_oN_nS_s$ with s = 1, 2 will be collectively referred to as organosulfur compounds including CHOS (n = 0) and CHONS (n = 1,2). The identified formulas containing isotopomers (i.e., $^{13}C$, $^{18}O$ or $^{34}S$) were not discussed. The double bond equivalent (DBE) is calculated using the equation: DBE = $(2c+2-h+n)/2$. Additionally, the modified index of aromaticity equivalent (Xc) was also calculated to estimate the degree of aromaticity, with the detailed data processing is presented in the Supplementary text (Yassine et al., 2014; Ye et al., 2020).

We assume that the different OSs may have similar ionization efficiency (Bateman et al., 2012), because the sulfate functional group on the OSs molecules are readily ionized during the ESI process and the ionization of OSs often takes place on the sulfate functional group (Lin et al., 2012). Based on this assumption and the fact that all the samples with similar carbon concentration were analysed in the same condition in this study (Jiang et al., 2021a)), the peak intensities of OSs ions could be compared to provide information on relative abundances among different samples by assuming that matrix effects were relatively constant in all samples (Lin et al., 2012; Kuang et al., 2016). However, the ionization efficiencies may vary among different OSs and lead to inconsistency between the ratios of peak intensities and the ratios of concentrations for other reasons, such as surface activity on ESI droplets (Kuang et al., 2016), but the sum-normalized peak intensities of the organosulfur compounds provide information on the relative abundances among different samples. To evaluate the associations between environmental variables and OrgSs compounds, we conducted non-metric multidimensional scaling (NMDS) analysis based on Bray−Curtis distances in R using the vegan package (Jiang et al., 2021a). From the NMDS analysis, the OrgSs compounds were dimensionally reduced to three components (NMDS1, NMDS2 and NMDS3) with stress values 0.09. The selected environmental parameters (Table S12) that have relationships or influences with/on the OrgSs composition were also fitted on the bitplots to evaluate the relationships between the distributions of OrgSs and environmental conditions, with *p*-values calculated over 999 permutations. The significant correlated factors were reserved and could be considered as the possible drivers that associated with molecular distribution. Score and loading plots were constructed according to NMDS variables from each OrgSs compound (gray dots and triangles). The potential drivers that associated with molecular distribution of OrgSs were indicated by arrows. Direction and included angle of arrow show the relationship between the driver and each dimension. Spearman correlation between the sum-normalized intensities of individual molecules and some important environmental variables/chemical tracers was performed in R, and then VK diagrams were plotted for each variable based on the Spearman correlation coefficients (Kellerman et al., 2014). Molecules found in at least 4 samples were adopted for correlation analysis. A false discovery rate-adjusted *p*-value was applied to avoid errors arising from using a large dataset.

## 3 Results and discussion

### 3.1 Abundance of sulfur-containing species

The annual average TS, inorganic sulfate-S and Org-S concentrations were 1.94±0.72, 1.31±0.60, and 0.62±0.26 µg/m$^3$, respectively (Table 1, n=40). The Org-S concentrations over Guangzhou were higher than those observed in a regional European site located in Hungary (0.02−0.33 µg/m$^3$) (Surratt et al., 2008; Luk´Acs et al., 2009), and close to the upper-bound measured in the U.S. (0.50 µg/m$^3$) (Table S1). These results suggest that the higher Org-S concentration in Guangzhou might be related to the high concentration of particulate matter and anthropogenic emissions. Furthermore, the high percentage Org-S content in fine particles (1.4±0.6%) was in the middle of the range estimated in the U.S. (0.75−2.0%), suggesting that Org-S might play a large role in the atmosphere and is probably an essential factor in the high particle pollution in Guangzhou compared to other sites. Our measurement of the annual Org-S to TS ratio was 0.33, which was significantly higher than that of ambient aerosols previously reported in Asia (0.01− 0.08) (Stone et al., 2012), the Arctic region (0.06) (Frossard et al., 2011), Hungary (0.06−0.20) (Luk´Acs et al., 2009; Surratt et al., 2008), and the U.S. (up to 0.22) (Chen et al., 2021). A study conducted in Germany estimated that up to 40% of the TS mass fraction can be contributed by organic molecules (Vogel et al., 2016), which is consistent with our results. There may be many reasons for the higher ratios in our measurements than at other sites, such as the high anthropogenic emissions, high relative humidity, or aerosol acidity levels, which were beneficial to the formation of organosulfur compounds (Bruggemann et al., 2020). Methanesulfonic acid (MSA) may account for a significant amount of the OrgSs mass in Guangzhou because it is a coastal city in southern China. The ratio of MSA-sulfur to Org-S was calculated based on the upper limit of the MSA-sulfur concentration (0.023 µg/m$^3$) measured in Hong Kong (a megacity near Guangzhou) during marine air mass influenced days (Huang et al., 2015). The estimated average ratio of MSA-sulfur to Org-S was 5.8±8.0, indicating that marine aerosols are probably also a non-ignorable source leading to the high Org-S values.

Table 1: Concentration (µg·m$^{−3}$) of sulfur-containing species and their fractionation in the PM$_{2.5}$ aerosols from Guangzhou (the samples with TS/SO$_4^{2−}$-S>2 or < 0.5, and Org-S/TS< $\delta$ $_{Org-S/TS}$ were excluded as described in section 2.1).

| Species/ratios | Spring (n=7) | Summer (n=13) | Autumn (n=5) | Winter (n=15) | Average (n=40) |
|---|---|---|---|---|---|
| TS | 1.92±0.38 | 1.57±0.68 | 1.97±0.97 | 2.25±0.64 | 1.94±0.72 |
| Sulfate-sulfur | 1.26±0.31 | 1.03±0.48 | 1.50±0.92 | 1.52±0.55 | 1.31±0.60 |
| Org-S | 0.66±0.19 | 0.54±0.28 | 0.47±0.27 | 0.72±0.21 | 0.62±0.26 |
| Sulfate-sulfur/TS | 0.66±0.09 | 0.67±0.14 | 0.74±0.11 | 0.66±0.10 | 0.67±0.12 |
| Org-S/TS | 0.34±0.09 | 0.33±0.14 | 0.26±0.11 | 0.34±0.10 | 0.33±0.12 |
| $f_{OS}$(%) | 48.2±15.9 | 45.4±21,9 | 30.9±14.5 | 39.1±18.9 | 41.7±19.7 |
| Org-S/OM (%) | 4.3±1.5 | 3.9±1.9 | 2.8±1.8 | 3.5±1.8 | 3.7±1.8 |

| Org-S/PM$_{2.5}$ (%) | 1.3±0.4 | 1.8±0.7 | 1.1±0.4 | 1.4±0.5 | 1.4±0.6 |

In this study, it was possible to estimate the fraction of OrgSs to the organic mass because the necessary mass-weighted average molecular weight (MW) of all OrgSs could be obtained from the FT-ICR MS analysis (Luk´Acs et al., 2009).

According to Tolocka and Turpin (2012), the fractional contribution of OSs to the organic mass ($f_{OS}$) can be estimated using the following equation:

$$f_{OS} = MW_{OS} \cdot \text{Org-S} / (MW_{Sulfur} \cdot \text{Organic Mass}) \qquad (1)$$

where $MW_{OS}$ and $MW_{Sulfur}$ denote the molecular weight of organosulfur compounds and S atom, respectively. The organic mass was derived from 1.8 times of the OC concentration measured by the Sunset OC/EC analyzer according to Tolocka and

190 Turpin (2012). In this study, the intensity-weighted average MW of OrgSs obtained from the FT-ICR MS analysis (see section 3.2) was used in the calculations. Our estimates of the OrgSs mass to organic mass ratio (41.7±19.7%) were comparable to observations of the organic mass in PM$_{10}$ over Hungary (Surratt et al., 2008; Luk´Acs et al., 2009), and the estimation at several sites for fine particulates (Frossard et al., 2011; Tolocka and Turpin, 2012), in which only OSs were considered (Table S1). Although there can be large uncertainties associated with this method, the estimates clearly showed

that OrgSs may be responsible for a sizable fraction of the ambient OM and PM mass, and it is essential to perform a detailed chemical characterization of OrgSs to improve our understanding of their sources, formation pathways, and fates in the ambient environment.

### 3.2 FT-ICR MS analysis of organosulfur compounds

In this study, a total of 15,998 organosulfur formulas were detected in the organic extracts of a yearlong sample set from the

200 FT-ICR MS analysis, and the organosulfur formulas detected in each sample accounted for an average of 33±4% of the total number of assigned molecules and 24−62% of the total MS intensity (mean: 44±8%). These compounds were distributed over a wide mass range. Based on the numbers of S and N atoms that appeared in each formula, these OrgSs could be grouped as CHOS$_1$, CHOS$_2$, CHON$_1$S and CHON$_2$S. The fractions of the four subgroups are listed in Table S2, with approximately 90% of the molecular number and 96% of the total MS intensity of OrgSs attributed to CHOS$_1$ and CHON$_1$S.

Because a sulfate group (−OSO$_3$H) carries four oxygen atoms and nitrooxy (-ONO$_2$) carries three O atoms, and they are all readily deprotonated in ESI, OrgSs with excess O atoms ($o/(4s+3n)\geq1$) are the likely organosulfates or nitrooxy-organosulfates (NOSs). However, other OrgSs (e.g., sulfonates), may also exist, but were not further considered. As many as 82−92% of the OrgSs detected in samples had $o/(4s+3n)\geq1$, suggesting that these compounds are potential OSs or NOSs, which is consistent with previous studies (Lin et al., 2012; Tao et al., 2014; Wang et al., 2019).

### 3.2.1 CHOS compounds

Table S2 summarizes the average characteristics (molecular weight, elemental ratios, and DBE) of the assigned CHOS and CHONS compounds. The majority (87−95%) of the CHOS formulas in the 55 samples contained enough O atoms to allow

for the assignment of −OSO$_3$H ($o/4s \geq 1$) in their formulas. The average intensity-weighted H/C, O/C, O/S and DBE values for the CHOS compounds were 1.77±0.03, 0.52±0.07, 6.7±0.4 and 2.77±0.20, respectively. The average H/C ratios of the CHOS compounds in this study were close to or higher than those previously reported in ambient aerosols (O'brien et al., 2014; Willoughby et al., 2014; Jiang et al., 2016; Jiang et al., 2020), clouds (Zhao et al., 2013; Bianco et al., 2018), and rainwater (Altieri et al., 2009) collected in different locations worldwide and analyzed by negative ESI-FT-ICR MS, indicating that the OrgSs in Guangzhou are enriched with saturated structures (Table S3). However, the average O/C ratios of the CHOS compounds identified in this study were slightly higher than those of cloud water (Bianco et al., 2018; Zhao et al., 2013), and comparable to the values measured in east-central Chinese cities (Wang et al., 2016; Wang et al., 2017a), but were much lower than those of CHOS compounds in polluted organic aerosols collected in Mainz and Chinese cities measured using high-resolution Orbitrap MS (Wang et al., 2019; Wang et al., 2021b). This implies that CHOS in Guangzhou might arise due to emissions from different sources and then be subjected to complex atmospheric oxidation processes. The differences identified from the comparisons also suggested that the CHOS compounds in Guangzhou might have a clear distinctive molecular composition compared to other locations due to the spatiotemporal heterogeneity, which suggests a need for further investigations of the sources and molecular distribution of OrgSs. The average DBE value of CHOS$_2$ compounds was approximately three times that of CHOS$_1$ compounds, indicating that CHOS$_2$ probably contains numerous aromatic OSs, but CHOS$_1$ compounds are dominated by OSs with long aliphatic carbon chains and low degrees of oxidation and unsaturation.

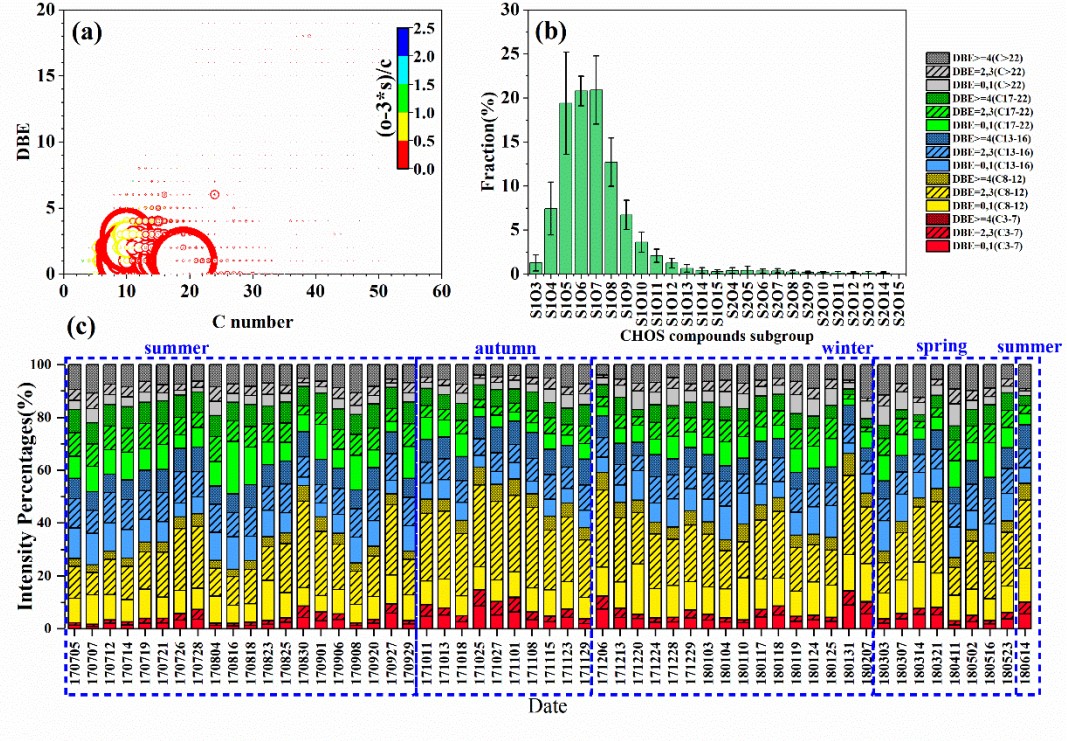

**Figure 1: Molecular distribution of CHOS compounds detected by FT-ICR MS for the sample set collected in Guangzhou. (a) Double bond equivalent (DBE) vs C number for all the CHOS compounds of all samples. Each circle denotes a molecule, and the colour bar and marker size denote the number of oxidation state and the average sum-normalized relative peak intensities of the compounds, respectively. Several most intense CHOS species list in descending order by their average intensities in Figure 1a are**

**$C_{19}H_{37}O_7S^-$, $C_{10}H_{19}O_5S^-$, $C_{10}H_{15}O_7S^-$, $C_{10}H_{17}O_7S^-$, $C_9H_{15}O_7S^-$, $C_{14}H_{27}O_5S^-$, $C_{15}H_{29}O_5S^-$; (b) Classification of CHOS species into different subgroups according to the numbers of S and O atoms in their molecules; (c) Percentages of signal intensity of each subgroup which divided based on the DBE value and the length of carbon skeleton in the formulas (all 55 samples were presented, yymmdd).**

Figure 1&S1 show the DBE, and C, and O atomic distributions in the CHOS compounds. The most abundant CHOS species

class identified in all our samples, had 5−7 O atoms and one S atom. The high number of O atoms in CHOS compounds probably suggested the existence of additional oxidized groups (e.g., hydroxyl and carbonyl). The CHOS compounds with a medium DBE value (=2, 3) accounted for the highest average percentages (40±5%) of the total MS intensity for the assigned CHOS compounds (Figure 1c). The additional double bonds (or olefinic structures) made them potential candidates for BVOC-derived OSs (Jiang et al., 2016; Lin et al., 2012). The CHOS compounds with DBE≤1 and DBE≥4, which were

tentatively assigned as saturated aliphatic-like and aromatic species, took up 34±6% and 26±2% of the total CHOS intensity, respectively. Note that the DBE-based criteria provided upper bound estimations of the relative abundance of aromatic OrgSs, which was about two times higher than that obtained using the aromaticity equivalent (Xc). The latter was considered a better index to describe potential monocyclic and polycyclic aromatic compounds with S atoms (Ye et al., 2020; Yassine et al., 2014). The aromatic OrgSs were dominated by phenyl OrgSs with Xc values between 2.500 and 2.7143, accounting for

76±9% of the total aromatic OrgSs peak intensity, possibly indicating important influences from anthropogenic primary emissions (Figure S1) (Song et al., 2018; Cui et al., 2019). The signal intensity of high-ring OSs (Xc≥2.7143) increased in winter and spring, suggesting the possibility of more combustion source emissions during these seasons.

Meanwhile, the low and medium DBE CHOS compounds (DBE<4) were further grouped based on the length of the C skeleton in the formulas to enable the distribution of BVOC-derived CHOS compounds to be studied. The relatively low

DBE (< 4) CHOS compounds with 3–7 carbons ($C_{3-7}$) were smaller compounds, which were probably the fragments produced by atmospheric oxidation processes or isoprene-derivatives (Nozière et al., 2010; Riva et al., 2016c; Rudzi´Nski et al., 2009). Larger compounds with $C_{>22}$ were also detected, but the average percentage of MS intensity to the total CHOS intensity was as small as that for $C_{3-7}$ compounds. The major fraction in low and medium DBE CHOS compounds (DBE≤3) was $C_{8-22}$ compounds, with $C_{8-12}$, $C_{13-16}$ and $C_{17-22}$ compounds accounting for 30±7%, 17±3% and 14±5% of the total OrgSs

intensity, respectively (Figure 1c). The $C_{8-22}$ compounds likely had associations with biogenic sources related to monoterpenoids/sesquiterpenoids and their dimeric oxidation products (Kristensen et al., 2016; Daellenbach et al., 2019). As highlighted by Kourtchev et al. (2016), the higher percentages of MS intensity for dimeric and trimeric BVOC oxidation products in both filed samples and laboratory-generated SOA could be related to the higher precursor and SOA mass. They suggested that a higher temperature could lead to an enhancement of oligomers because it affects not only the biogenic

emissions but also the partitioning of dimeric and monomeric compounds in the gas and particle phases. In this study, the average temperature during the sampling period was 24 ℃. According to Kourtchev et al. (2016), the average maximum temperature of $24\pm6$ ℃ could have an oligomer fraction of 0.3 among the total intensity of all peaks in the mass spectrum. This higher percentage of MS intensity suggested the importance of dimeric oxidation products to the aerosols. However, it should be noted that $C_{8-22}$ CHOS compounds have also been reported in previous studies and are proposed to be mainly

derived from the photooxidation of long-chain alkanes from vehicle emissions (Tao et al., 2014; Riva et al., 2016b), and the reactions of $SO_2$ and unsaturated acids in ambient particle samples (Shang et al., 2016; Zhu et al., 2019). For example, compounds such as $C_6H_{11}O_6S^-$, $C_7H_{13}O_6S^-$, $C_8H_{17}O_6S^-$, and $C_{10}H_{19}O_6S^-$ were observed in both the formation processes via monoterpene ozonolysis intermediates (Ye et al., 2018) and uptake of $SO_2$ by olefinic acid (the possible olefinic acid precursors were all detected in the FT-ICR MS analysis) (Zhu et al., 2019). Therefore, due to our limited data, the origins of

CHOS with a low DBE remains large uncertainties and needs to be confirmed by further studies.

### 3.2.2 CHONS compounds

As shown in Table S2, the assigned CHONS formulas in each sample accounted for 27−42% and 16−41% of the OrgSs in terms of the number of formulas and MS intensity, respectively. These compounds had a higher average MW, O/C, O/S, and DBE value than the CHOS compounds, which was probably due to the presence of additional nitrate groups. The results of

the comparison between the average H/C and O/C ratios of the CHONS compounds and those reported previously were consistent with the results for the CHOS compounds (Table S4). Despite CHONS compounds containing two N atoms also being identified, their relatively low MS intensity makes them less important than those containing one N atom. In this study, 70-89% (in number) of the CHONS compounds had $o/(4s+3n)\geq1$, implying that they were candidates for NOSs. It has been demonstrated that NOSs can form via the photooxidation of BVOCs in smog chamber experiments conducted under high

$NO_x$ conditions (Surratt et al., 2008; Iinuma et al., 2007b). However, recent combustion experiments have found that freshly emitted organic aerosols also contain a significant fraction of CHONS compounds, especially in coal combustion aerosols (Song et al., 2018; Blair et al., 2017; Tang et al., 2020; Cui et al., 2019).

The CHONS species observed in this study were $O_4N_1S_1-O_{15}N_1S_1$ and $O_7N_2S_1-O_{14}N_2S_1$ class species, of which the $O_7N_1S_1$ class species was the most abundant family. The most abundant chemical formula in most samples was $C_{10}H_{16}NO_7S^-$ with

DBE=3 and m/z = 294.0653, which is usually considered to be generated from the oxidation of α-pinene in the atmosphere (Figure S2a) (Surratt et al., 2008). However, it was also identified in coal combustion-emitted aerosols in a recent study, indicating that this compound probably had multiple sources (Song et al., 2018). The distribution of the CHONS compounds across DBE and C numbers was quite similar to that of the CHOS compounds (Figure S2a). From the equation of the DBE calculation, each nitrooxy group in the CHONS compounds also contained one double bond and therefore contributeed to a

DBE value of 1. Therefore, the DBE value minus the number of N atoms (DBE−N) is a better criterion to determine the aromatic structure or whether this is possible (Lin et al., 2012). The CHONS compounds were dominated by olefinics ((DBE−N)=2, 3), followed by saturated aliphatic ((DBE-N)≤1) and aromatic ((DBE−N)≥4) CHONS (Figure S2c&d).

Furthermore, the most abundant classes in the saturated aliphatic and olefinic CHONS were $C_8$-$C_{12}$ compounds with O numbers higher than 7 (Figure S2 b&c&d).

### 3.3 Comparison and potential precursor apportionment of OrgSs

A substantial overlap of OrgSs were observed in this work with source samples, including BBOA, coal combustion organic aerosols (CCOA) and vehicle emissions, non-road excavator and ship emissions, and tunnel aerosol samples (Tang et al., 2020; Cui et al., 2019). Figure 2a shows a comparison of the molecular characteristics of OrgSs for our field samples and source samples. The intense OrgSs in Guangzhou were mainly composed of unsaturated aliphatic molecules, which was similar to the tunnel aerosol sample that may have undergone atmospheric aging processes. However, the OrgSs in fresh vehicle emissions were abundant in aromatics, with 69% of identified OrgSs having Xc≥2.500 (Table S5). Despite the diesel fuel combustion-emitted aerosols also containing unsaturated aliphatic molecules with a high intensity, their oxidation levels were lower than those of our field samples. Both BBOA and CCOA were abundant with aromatic and highly unsaturated organosulfur molecules, which had distinctive molecular characteristics compared to our field samples. Although 50±5% (in number) of the OrgSs identified in Guangzhou could be attributed to aromatic OrgSs, most of them had a low intensity. Although combustion sources can emit large numbers of OrgSs, the primary low-oxidative and aromatic OrgSs abundant in source samples had a low MS intensity in our ambient samples. This probably suggested that the OrgSs in Guangzhou were less or indirectly affected by primary emissions (e.g., secondary formation via combustion-emitted precursors).

Additionally, we apportioned the detected OrgSs into five groups based on their potential precursors, including BVOC-derived OSs (e.g., isoprene-derived OSs, monoterpene-derived OSs, and other BVOC-derived OSs from the precursors of green leaf volatiles), anthropogenic VOCs-derived OSs from the precursors of aromatics and anthropogenically emitted alkane precursors, and multiple-source-derived OSs from carbonyl compounds, unsaturated acid, and alkanes. Details of these OSs formulas with the determined precursors are listed in Table S6-10. The OSs that were identical to the published OSs (their precursors previously have been verified) were temporarily considered to have the same precursors as the published OSs in the this study. This method has been widely used because its feasibility is based on the high mass resolution of HR-MS for the identification of mass peaks in conjunction, with the assignment of formulas using a narrow mass tolerance (Lin et al., 2012; Kuang et al., 2016; Ye et al., 2020).

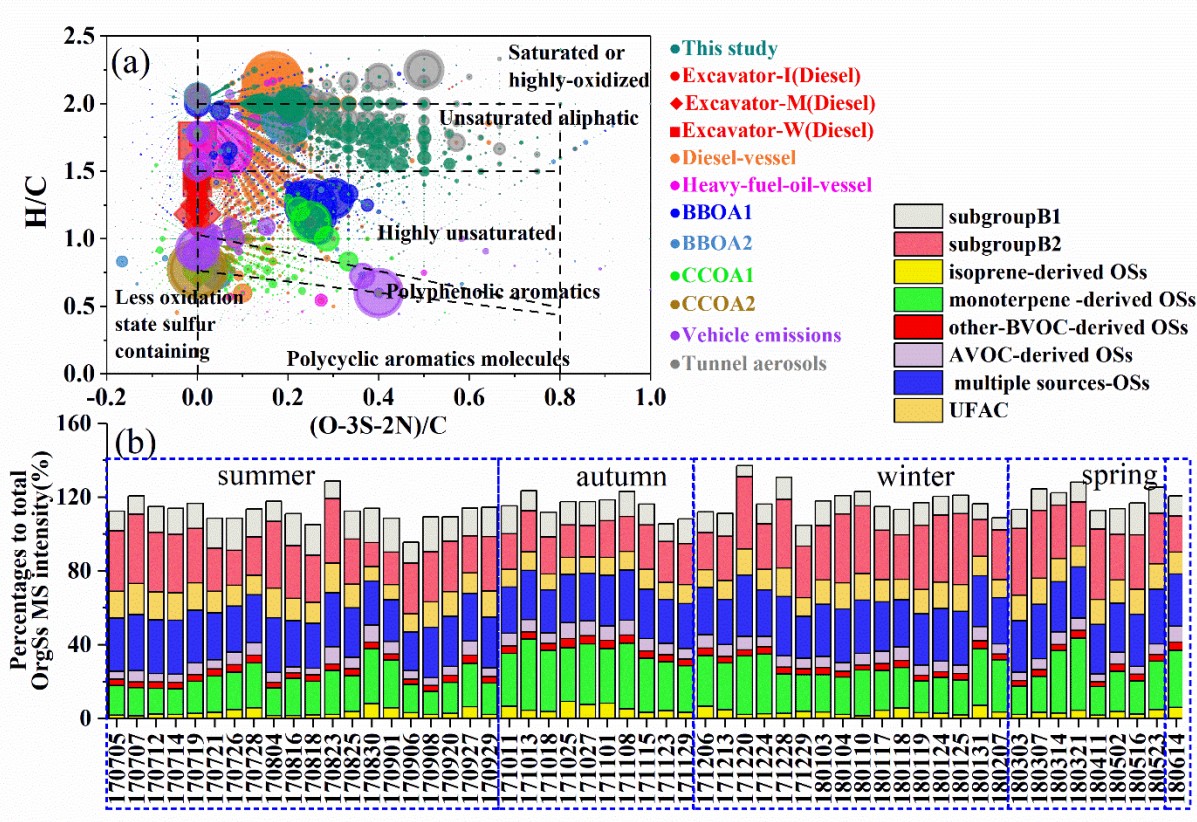

**Figure 2: (a)** Van Krevelen diagrams of the field samples collected in Guangzhou and source samples obtained from Cui et al. (2019) and Tang et al. (2020), including biomass burning organic aerosols (BBOA), coal combustion organic aerosols (CCOA), vehicle emissions, tunnel aerosols, and off-road engine emissions (excavator and vessel). Excavator-I, -M and -W denotes the operation modes of idling, moving, and working, respectively. The marker size denotes the percentages of MS intensity to the total identified organosulfur compounds. **(b)** Annual variations of potential precursor of detected OSs to the total identified organosulfur compounds MS intensity ; subgroupB1 denotes OSs having C>8, DBE<3 and 3<O<7 (for CHOS)/ 6<O<10 (for CHONS), while subgroupB2 denotes OSs having C>8, DBE<3 and O≥7 (for CHOS)/ O≥10 (for CHONS).

Figure 2b shows the annual variations of the total MS intensity of the five OSs groups as a percentage of the total OrgSs MS intensity, with annual average proportions of 3.8±1.9%, 23±6.7%, 3.6±0.5%, 6.1±1.4% and 27±2.3% for isoprene-derived OSs, monoterpene-derived OSs, other BVOC-derived OSs, anthropogenic VOC-derived OSs and multiple-source-derived OSs, respectively. The high percentages of MS intensity for known terpene-derived OSs to the total OrgSs intensity in this study was consistent with previous observations of the dominance of terpene-derived OSs in Guangzhou (Wang et al., 2017b; He et al., 2014; Bryant et al., 2021). Several highly abundant formulas of terpene-derived OSs, $C_{10}H_{16}O_7NS^-$ (m/z 294); $C_{10}H_{19}O_5S^-$ (m/z 251), $C_{10}H_{15}O_7S^-$ (m/z 279), $C_{10}H_{17}O_7S^-$ (m/z 281) and $C_9H_{15}O_7S^-$ (m/z 267), have been widely reported as

being predominantly formed by the acid-catalyzed chemistry of BVOC-derived oxidation products (Hettiyadura et al., 2019; Bruggemann et al., 2020). Notably, $C_9H_{15}O_7S^-$ was also observed as a secondary product formed by isoprene (Meade et al., 2016), which was partially supported here by the positive correlation between their sum-normalized intensity and the concentration of MTLs (SOA tracers of isoprene, the sum of 2-methylthreitol and 2-methylerythritol) ($r=0.73$, $p<0.01$) (Li et al., 2013). Isomers acting as both anthropogenic and biogenic precursors cannot be distinguished by an FT-ICR MS analysis, because compounds with specific m/z values are manifested as a single signal in the FT-ICR mass spectra, and our reported ratios may therefore be subjected to uncertainty. Furthermore, due to the limitation of detection techniques and trace concentrations, the incomplete OSs list in the SI for the different SOA precursor groups may also leads to uncertainty in our classification.

Polycyclic aromatic hydrocarbons have been recognized as precursors of aromatic OSs from laboratory evidence (Riva et al., 2015). Aromatic OSs with benzyl and polycyclic aromatic C backbones, such as $C_6H_5SO_4^-$, $C_7H_5SO_4^-$, $C_7H_7SO_4^-$, $C_8H_7SO_4^-$, and $C_9H_{11}SO_4^-$ and several OSs from the photooxidation of naphthalene and 2-methylnaphthalene, have been widely observed in urban and semirural fine particles worldwide (Le Breton et al., 2018; Huang et al., 2018a; Wang et al., 2018; Hettiyadura et al., 2015; Bruggemann et al., 2020) and were also detected in our samples. However, presently only a few species of aromatic OSs with a relatively low MS intensity have been classified. Aromatic OrgSs with Xc≥2.5 accounted for 9−20% of the total OrgSs peak intensity in this study, emphasizing the significant contribution of anthropogenic emissions in Guangzhou.

Among the classified OrgSs with their precursors from multiple sources, a high intensity fraction that was likely derived from unsaturated fatty acids (USFA) was identified, and contributed 8%−17% (average: 12%) of the total OrgSs potentially assigned, despite the limitations imposed by the large numbers of different OrgSs variants. We observed a positive correlation between USFA-derived OSs and RH ($r^2=0.19$, $p<0.01$), which partly supported the mechanism of USFA-derived OSs formation by direct $SO_2$ uptake. This was consistent with a recent study showing that USFA-derived OSs accounted for a high fraction of the total OSs intensity (5%−7% sulfur of all the OrgSs) and were positively correlated with RH in the PRD (Zhu et al., 2019). The authors tentatively attributed the formation of these OSs to the direct reaction of $SO_2$ with unsaturated acids in ambient particle samples in the presence of gas-phase oxidants such as OH radicals or $O_3$, because several laboratory studies (Shang et al., 2016; Passananti et al., 2016) have observed a dependency of USFA-derived OSs formation on RH. It has been suggested that RH is an important influencing factor, and increasing humidity would accelerate $SO_2$ uptake and thereby OSs formation.

We noted that the subgroup of OSs with unidentified precursors and C>8, DBE<3, and 3<O<7 (for CHOS)/ 6<O<10 (for CHONS) accounted for 27±7% of the MS intensity of the total identified OrgSs. This subgroup of OSs (subgroupB1) is characterized by a high molecular weight, alkyl chains and a low degree of oxidation, and was first reported by Tao et al. (2014) who speculated that the precursors of this subgroup of OSs could be long-chain alkanes from traffic emissions. The long-chain alkanes were photooxidized by a mixture of oxidants under typical urban conditions and formed hydroxylated or carbonylated products, which were further esterified to form alkyl OSs. Riva et al. (2016a) conducted an experiment on the

photooxidation of alkanes in an outdoor smog chamber and proposed that gaseous epoxide precursors with subsequent acid-catalyzed reactive uptake onto sulfate aerosols and/or heterogeneous reactions of hydroperoxides can also be used to explain

the formation of alkane-derived OSs. Furthermore, the formation of OSs via heterogeneous reactions of $SO_2$ with USFA was also important for these highly saturated OSs (Zhu et al., 2019). The total relative intensity of subgroupB1 was positively correlated with RH and the concentrations of chemical tracers associated with fossil fuel combustion ($Cl^-$, steranes and hopanes: $\Sigma SH$) (Figure S3), support the influences of heterogeneous reactions and photooxidation of traffic-emitted long-chain alkanes on subgroupB1, but more detailed source information is required to confirm this.

**3.4 Possible formation pathways of OrgSs and the influencing factors**

As shown in the previous section, OrgSs in the atmosphere in Guangzhou were significantly influenced by different sources, including both primary emissions and secondary formation. However, although a variety of reaction pathways have been proposed for the secondary formation of OSs, the formation mechanisms of OSs in the atmosphere are not fully understood. Bruggemann et al. (2020) reviewed and summarized the OSs formation pathways that have been identified thus far and

outlined their potential atmospheric relevance. It has been shown to be kinetically feasible for acid-catalyzed reactions of the epoxides formed by the oxidation of VOCs to produce Oss, and this mechanism has been widely adopted to explain OSs formation (Surratt et al., 2007; Iinuma et al., 2007b; Surratt et al., 2008; Surratt et al., 2010; Lin et al., 2013). The distribution of OS products is expected to depend on precursor concentrations (including organic compounds and anthropogenic pollutants, e.g., $NO_x$ and $SO_2$), acidity, RH, and oxidant concentrations. A recent study conducted in South

China also revealed that high levels of isoprene-derived OSs were derived from the acid ring-opening reactions of isoprene-derived epoxydiols (He et al., 2018). In view of the products' molecular structure, the acid-catalyzed ring-opening of epoxides by the addition of inorganic sulfate ions usually leads to the formation of β-hydroxyl OSs (Figure 3, Scheme 1) (Lin et al., 2012). Thus, the OSs and NOSs generated from the epoxide pathway usually have O>4 for CHOS compounds and O>7 for CHONS compounds, respectively. Lin et al. (2012) removed $-SO_3$ from the OrgSs to obtain the corresponding

alcohols and examined their presence by comparing them with the non-S-containing formulas in the samples collected at the PRD. They found that 65−75% of the CHOS compounds could be formed from the epoxide intermediate pathway. In our samples, an upper bound estimation for the fraction of OrgSs formed via the epoxide intermediate pathway could reach half number of the detected OrgSs because 46±12% (both in number and MS intensity) of OrgSs satisfied the above criterion (Table S11). The percentage of MS intensity for these OrgSs had a decreasing trend from summer to winter, and then

increased in spring. It presented positive correlations with the fraction of $SO_4^{2-}$ in secondary ion aerosols (SIA) ($r$=0.54, $p$<0.01), temperature ($r$=0.63, $p$<0.01) and biogenic SOA tracer ($r$=0.34, $p$<0.05), which was consistent with a recent study (Bryant et al., 2021) and suggested that the temperature and available particulate $SO_4^{2-}$ are important influencing factors in the formation of OrgSs via the acid-catalyzed ring-opening of epoxides.

(a) Scheme1: Acid-catalyzed epoxides ring-opening pathway

(b) Scheme2:Heterogeneous oxidations with SO$_2$ uptake pathway

(c) Scheme3: One of possible NOS formation pathway

**Figure 3: The two potentially important OSs formation mechanisms in Guangzhou (Duporte et al., 2020; Ye et al., 2018; Bruggemann et al., 2020; Aoki et al., 2020; Lind et al., 1987). (a) Proposed OSs formation mechanism of acid-catalyzed ring-opening of epoxides; (b) Proposed OSs formation mechanism for heterogeneous reactions of SO$_2$ and the secondary products from ozonolysis unsaturated hydrocarbon at high relative humidity; (c) one of possible NOSs formation pathway.**

From the Org-S mass data, as shown in Table 1, the Org-S, along with TS and sulfate-sulfur levels exhibited a clear seasonal variation, with all having higher values in autumn and winter than in spring and summer (ANOVA, $p<0.01$). The higher levels of sulfur-containing species in cold seasons may be due to the higher anthropogenic emissions. However, both the Org-S/PM$_{2.5}$ and $f_{OS}$ exhibited different seasonal variation, with higher ratios observed in summer than in the cold seasons. This different seasonal characteristic may have been influenced by several factors, including precursor emissions of BVOCs, and high RH levels, which might increase the SO$_2$ uptake and formation of OrgSs during warm seasons (Bruggemann et al.,

2020; Zhu et al., 2019). Additionally, gas-phase oxidation initiated by $O_3$ or OH radicals, which promote the generation of oxidation products, hydroxyl, and carbonyl (Riva et al., 2016b), also contributed to the formation of OrgSs. This was supported by the finding that the Org-S concentration was positively correlated with oxidant levels (indicated by $NO_x+O_3$, $r=0.40$, $p<0.01$) and $SO_2$ ($r=0.37$, $p<0.05$) (Figure S4). Furthermore, we observed that the Org-S concentration was positively correlated with $NO_3^-/SIA$ ($r=0.41$, $p<0.01$), but negatively correlated with the $SO_4^{2-}/SIA$ ratio ($r=-0.40$, $p<0.01$), probably suggesting the presence of competition between $SO_4^{2-}$ and OrgSs in their formation (Figure S4). This was inconsistent with a previous observation that OSs increased with $SO_4^{2-}/SIA$, which showed a linear relationship with particulate acid (Guo et al., 2016; Wang et al., 2018). Several studies have also reported that some isoprene-derived OSs, which were produced through the reactive uptake of isoprene-epoxydiol (IEPOX) onto acidic particles, exhibited no correlation with aerosol acidity (He et al., 2014; Lin et al., 2013; Worton et al., 2013). In this study, the pH of all samples was below 5 and we did not observe a significant correlation between pH values (or $H^+$) and the Org-S concentration, but a molecular-level assessment showed that a small number of individual organosulfur species were significantly correlated with the $H^+$ concentration, probably indicating that the variation in particulate acid have minor associations with OrgSs formation in overall. Additionally, we found that the Org-S concentration had a non-significant correlation with levoglucosan and $\Sigma SH$ concentration, indicating that primary biomass burning and fossil fuel combustion probably had little or no direct impact on the variation of Org-S, which was consistent with the comparative analysis reported in section 3.3.

Our findings also provide support for the heterogeneous reactions of the $SO_2$ uptake pathway, which was expected because, as discussed above, the Org-S concentration was positively correlated with $O_3$, $NO_2$, and $SO_2$, and RH was negatively correlated with $SO_2$ (Ye et al., 2018; Bruggemann et al., 2020). Both laboratory studies and field observations have suggested that $SO_2$ uptake by unsaturated compounds and naphthalene, and the formation of OSs were shown to increase with higher RH levels (Zhu et al., 2019; Shang et al., 2016; Riva et al., 2015). Blair et al. (2017) also reported an increase in concentration with increasing RH for some specific aromatic OSs in biodiesel and diesel fuel SOA. Ye et al. (2018) found that $SO_2$ uptake and OSs formation increased with higher RH levels for the monoterpene ozonolysis intermediate, which was likely due to reactions between $SO_2$ and organic peroxides. Given the high RH levels during the sampling campaign (average=$70\pm14\%$) and the above results, it was reasonable to speculate that $SO_2$ was preferentially partitioned into the aqueous phase and formed $HSO_3^-$, with the formation of OSs through reactions between $HSO_3^-$ and the organic precursor ozonolysis intermediate, organic (hydro-)peroxides (Figure 3, Scheme 2) (Ye et al., 2018; Bruggemann et al., 2020).

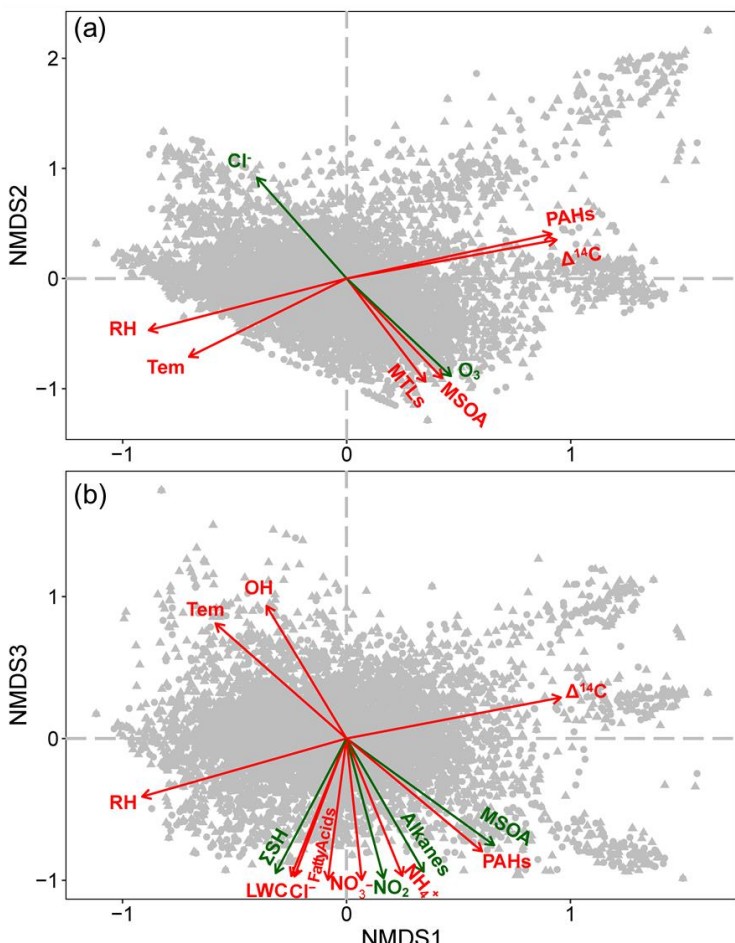

**Figure 4: Nonmetric multidimensional scaling analysis of the influences from environmental parameters on organosulfur compounds. The three-dimensional ordination Ordinations are based on Bray−Curtis (stress = 0.09, non-liner $r^2$=0.99), which utilizes sum-normalized relative compound intensity. Environmental parameters listed in Table S12 were fit to the ordination. Gray-shaded dots and triangles are CHOS and CHONS compounds, respectively. Variables with significance levels of <0.05 (green) and <0.01 (red) are shown, and nonsignificant correlations are not shown.**

To support our speculation and discern the possible environmental drivers of the molecular distribution of OrgSs, NMDS analysis of OrgSs was conducted (Figure 4 and Table S12). Among the significant drivers, it was noted that RH was important and associated with the seasonal distribution of the OrgSs composition, with RH and temperature clustered at the negative end of the first dimension, while $\Delta^{14}$C was positively correlated with the first dimension. Notably, an "older" $^{14}$C age of organic carbon was generally accompanied by a high RH, and the results a from recent compound-specific dual-carbon isotopic ($\delta^{13}$C and $\Delta^{14}$C) analysis of dicarboxylic acids (SOA tracers) indicated that large fractions of the organic

mass were substantially supplied by the aqueous-phase transformation of fossil-fuel precursors (Xu et al., 2021a). These results indicate the importance of the aqueous-phase formation of OrgSs via fossil-fuel precursors in addition to the direct emissions from combustion sources (Wang et al., 2021a).

Additionally, we found that the BVOC-derived SOA tracers and $O_3$ were distributed at the negative end of the second dimension, while the anthropogenic species (e.g., $NO_3^-$, $NH_4^+$, $NO_2$, fatty acids, and $\Sigma$SH) and aerosol liquid water content (LWC) were negatively correlated with the third dimension, with the opposite pattern for temperature and OH radical (Figure 4). This probably suggested that there were the different oxidation processes involved in the formation of OrgSs between the warm and cold seasons, with cold seasons often experiencing high anthropogenic emissions, while high biogenic emissions occur in warm seasons (see Supplementary text). The cluster of BVOC-derived SOA tracers and $O_3$ probably suggested that SOA products produced by the reactions of BVOCs with $O_3$ were important precursors of the OrgSs in this study, which was supported by recent studies showing that daytime/night-time $O_3$-related oxidation in the presence of $SO_2$ also potentially contributed to the OSs formation (Xu et al., 2021b; Chen et al., 2020). However, the cluster of anthropogenic organic compounds, together with reactive nitrogen species and LWC, probably also suggested the influence of aqueous-phase reactions of fatty acids and other fossil-fuel precursors on OrgSs formation, particularly the inorganic nitrogen species-related formation of NOSs (Bryant et al., 2021). This was expected because aerosol LWC provides a medium for aqueous-phase reactions (Guo et al., 2016; Liu et al., 2017; Wang et al., 2018), and positive correlations were observed between LWC and secondary inorganic aerosols ($r=0.69$, $p<0.01$), particularly the inorganic nitrogen species. Moreover, a direct assessment of the relationships between individual compounds and LWC, $NO_3^-$, and $NH_4^+$ suggested that an increase in their concentrations would promote the formation of CHONS species. It was found that 100%, 64%, and 74% of the OrgSs that had positive correlations ($p$-adjusted with "fdr") with the LWC, $NO_3^-$, and $NH_4^+$, respectively, were CHONS species (Table S13). This further indicated that OrgSs formation via aqueous-phase chemistry in Guangzhou was influenced by LWC, such as the $NO_3$-initiated oxidation and acid-catalysed epoxide pathways (Wang et al., 2020; Xu et al., 2021b). Recently, Bryant et al. (2021) reported that oxidants and temperature are important factors that affect OSs formation in Guangzhou, and high-$NO_x$ pathways became more important in the winter when anthropogenic emissions usually high, whereas low-$NO_x$ formation pathways were dominant in summer. The observed opposite influence of OH radicals and inorganic species on OrgSs distributions also suggested OrgSs formation occurred through heterogeneous OH radical oxidation when anthropogenic emissions were low (Chen et al., 2020; Lam et al., 2019). These results suggested the importance of atmospheric oxidation on the molecular composition of OrgSs, but there may be distinct effects for different oxidation processes (i.e., gas-phase $O_3$ oxidation, liquid-phase $NO_3$-initiated oxidation and heterogeneous OH radical oxidation).

## 4 Conclusions

This study investigated the abundance and molecular characteristics of the atmospheric organic sulfur fraction in Guangzhou,

South China, with yearly PM$_{2.5}$ samples collected and analyzed. The results showed that organosulfur can accounted for up to 42% of the total organic mass on average, and is particularly important in fine particulate pollution. A molecular composition analysis performed using negative ESI-FT-ICR MS suggested a complex chemical composition and multiple sources. The substantial overlap of the organosulfur species observed in this study with those identified in previous chamber and field studies suggested that alternative mechanisms of organosulfur formation could be important in the atmosphere over Guangzhou. We also compared the organosulfur species composition with several source samples and found clear differences among different source samples. Many organosulfur species in our data that were previously classified as having biogenic, anthropogenic, or unidentified sources were also found among the collected source samples. Despite most of time the aromatic organosulfur compounds had a relatively low MS intensity, the high fraction of formular number to the total assigned OrgSs suggesting that extensive human activities and high level of anthropogenic emissions (e.g., vehicle emissions, coal combustion and biomass burning) made an important contribution to the OrgSs composition.

Because the formation pathways and influencing factors of OrgSs were hardly recognized, we employed an NMDS analysis based on the large amounts of data obtained from the FT-ICR MS analysis and chemical tracers. Both the mass concentration and chemical composition data indicated the potential OrgSs formation from acid-catalyzed aqueous-phase reactions, and RH and oxidant levels (NO$_x$+O$_3$) were important environmental drivers that influenced the OrgSs distributions and heterogeneous reactions of SO$_2$ uptake in OrgSs formation. This was consistent with most previous observations of higher yields of organosulfur species at elevated RH during laboratory experiments. The oxidation of BVOCs with O$_3$ and oxidation of anthropogenic VOCs in the presence of NO$_x$ were two potentially important pathways for the formation of OrgSs or their precursors. From our results, we stressed that although RH was an immutable parameter, reducing SO$_2$ emissions alone was insufficient to decrease the OrgSs fraction in atmospheric particulates, and it was also necessary to reduce NO$_2$ and other anthropogenic emissions.

**Data availability**

Data are available upon request, by the corresponding authors.

**Author contributions**

HJ and JL designed the experiment. HJ, JT, BJ and YL carried out the measurements. HJ, JT and YM analysed the data. HJ, JL and GZ organized and supported the samplings. JL and GZ supervised the study and worked for funding acquisition. MC and JT provide the original data about the source samples. HJ wrote the paper. JL, GZ, MC, YM, SZ, XZ, and GZ reviewed and commented on the paper.

**Competing interests**

The authors declare that they have no conflict of interest.

**Acknowledgements**

This study was supported by the Natural Science Foundation of China (41773120, 42192514, and 41977177), National Key R&D Program of China (2018YFC1802801), the, Guangdong Foundation for Program of Science and Technology Research (Grant No. 2019B121205006 and 2020B1212060053).

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
