# Peer review of "Molecular Characteristics, Sources, and Formation Pathways of Organosulfur Compounds in Ambient Aerosol in Guangzhou, South China"

_Atmospheric Chemistry and Physics, 2021_

## Referee Comment (RC1)

**General Comments**

The authors presented a work that summarizes a yearlong measurement of organosulfur compounds (OrgSs) in $PM_{2.5}$ collected in Guangzhou. The authors carried out detailed characterization of the abundance and composition of OrgSs using high-resolution mass spectrometry. The authors further examined the molecular characteristics of the detected OrgSs based on the elemental compositions. The association of OrgSs with other chemical tracers and meteorological data were assessed to understand possible sources, pathways, and governing factors that explain their presence in Guangzhou based on current mechanistic knowledge gained through prior laboratory and field studies. The manuscript includes a considerably large dataset that the authors meticulously collected, examined, and compiled. My main concerns about this manuscript are related to the clarity of its presentation. Detailed comments are as follows.

**Specific Comments**

1. Consistency of the main message: The abstract is highlighting epoxide chemistry, $SO_2$ uptake and heterogeneous oxidation. However, the heterogenous reaction involving $SO_2$ seems to get neglected in the last paragraph of the introduction. The heterogeneous oxidation was mentioned there but this point was not even given enough discussion and not backed up by convincing evidence in the manuscript (Line 411-413). The authors are urged to carefully pick their main points and make sure they are consistently articulated throughout the manuscript.

2. Section 2.1: I think the authors should provide some necessary discussion regarding the sampling site and its connection to the emission sources. This will provide some contexts for the comparison of the samples in this study with the source samples discussed in Section 3.3. Now it gives the readers impression that the source samples are suddenly brought up from nowhere.

3. Line 97–98: I was confused by the criteria of excluding samples described here: "In this study, **the TS to sulfate-sulfur ratios of samples greater than 2 or less than 0.5**, which were considered as a measure of gross measurement error, were **also** excluded from further analysis." It seems to me that "TS to sulfate-sulfur ratios of samples greater than 2 or less than 0.5" reflects a rather loose tolerance when compared to previous studies such as Shakya and Peltier, 2015. However, "also" indicates there were other criteria. Did the authors miss something here? I was also confused by the total number of samples analyzed. It was stated here that 40 samples were reserved but later authors said a total of 55 samples were analyzed by ESI-FT-ICR MS (Line 105). How many samples are the presented data (particularly those in Table 1) representative of? Please clarify.

4. How were $PM_{2.5}$ and OM determined for calculating the fractions summarized in Table 1? Is OrgSs/OM in Table 1 same to $f_{OS}$ defined in Line 154?

5. Figure 1c: The x tick labels are roughly spaced by 3 months but the number of samples between the tick labels are not the same and didn't match the number of samples for each season (Table 1), which seems arbitrary to me. The authors may consider spacing out the

tick labels based on seasons. Please also clarify the starting and ending months for the first and last sample collected in each season in Table 1 either as a note associated with the table or in the methods section.

6. In Table 1: Why OrgSs/OM >>1. Is the unit %? If so, the fact that OrgSs/OM values were much greater than Org-S/OM is problematic. The authors should discuss possible reasons in the context of measurement uncertainties and assumptions made in the quantitative calculations.

7. Line 239-241: By definition, aliphatic compounds include both saturated and unsaturated compounds ((DBE − N) < 4). Therefore, aliphatic is not the most accurate term to refer to compounds with (DBE − N) <= 1. In addition, it is unclear to me how one can derive what was stated (most abundant classes in aliphatic CHNOS being $C_{8-12}$ with O numbers > 7) from Figure S2e&f. Wrong referenced figures?

8. Line 317-319: Why did the authors show the correlations with Cl-, steranes and hopanes while the preceding discussion was primarily about the long chain alkanes? This seems out of context. Please elaborate.

9. Line 330-331 and Scheme 1: $SO_4^{2-}$ should play a main role in the reactions with epoxides explaining OS formation under relevant atmospheric conditions as $HSO_4^-$ is a much weaker nucleophile (Aoki et al., 2020).

10. Line 335-337: The authors showed that there is a considerable number of compounds that could be possibly explained by epoxide pathway. What is the summed relative abundance of these compounds to total OrgSs? This is one of the highlights of the paper and there is so much potential for further discussion later in the manuscript. I would expect to see if they vary seasonally and how they correlate with other chemical tracers and environmental parameters such as $SO_4^{2-}$, pH, RH, LWC, etc.

11. Line 344-345: I don't think this statement was accurate according to what was shown in Table 1. Except for sulfate-sulfur, Org-S and TS mean values are both higher in Spring than in Autumn. Additional analysis is recommended to determine whether there are any statistically significant differences between the four seasonal averages.

12. Line 348: cooler instead of warm seasons? In fact, all three other seasons?

13. Line 404-406: The numbers (72%, 65%, and 75%) cited to support the statement don't agree with Table S13. They should be 100%, 64%, and 74%, respectively. Are there errors in Table S13 or the text? Table S13 also appears to be incomplete. For example, the last three cells in the row of MTLs are missing percentage data.

14. Line 416: 25% is not consistent with Table 1 which shows an average OrgSs/OM of 13.9%.

**Technical Corrections**

1. The title: replacing "Drivers" with Drive makes more sense to me.
2. Please make sure the numbers in all chemical formulas are subscripted throughout the manuscript.
3. Line 247: I think the authors intended to say Figure 2a instead of Figure S2a.
4. Line 303: I know PRD stands for Pearl River Delta but the abbreviation was not defined in the text. In a few other occasions, Pearl River Delta was used but not abbreviated.
5. Line 353: Was reference to Figure S4 supposed to be here?

6. Figure 4: In the figure caption, Table S6 doesn't seem to be the correct reference for environmental parameters.
7. The reference to Figure S1 is nowhere to be found in the text. Is Figure S1 complementary to Figure 1, which only show the subgroup CHOS?
8. Tables S1-S4: The cell information could be better aligned. Consider adding cell outlines to guide the eyes.
9. Table S2: What does SOC standard for? Sulfur containing organics? What is SOC formulas set? Please clarify and make sure they are defined and explained in the revised manuscript.
10. Table S6-S8: Please subscript atomic numbers in the chemical formulas

**References**

1. Aoki, E., Sarrimanolis, J. N., Lyon, S. A. & Elrod, M. J. Determining the Relative Reactivity of Sulfate, Bisulfate, and Organosulfates with Epoxides on Secondary Organic Aerosol. *Acs Earth Space Chem* (2020) doi:10.1021/acsearthspacechem.0c00178.